# Dietary Habits of Patients with Coronary Artery Disease: A Case-Control Study from Pakistan

**DOI:** 10.3390/ijerph19148635

**Published:** 2022-07-15

**Authors:** Muhammad Kamran Hanif, Yahui Fan, Lina Wang, Hong Jiang, Zhaofang Li, Mei Ma, Le Ma, Mao Ma

**Affiliations:** 1The First Affiliated Hospital, Xi’an Jiaotong University Health Science Center, Xi’an 710061, China; muhammad.hanif@uipt.uol.edu.pk; 2School of Public Health, Xi’an Jiaotong University Health Science Center, Xi’an 710061, China; fyh14042166@stu.xjtu.edu.cn (Y.F.); wln305@stu.xjtu.edu.cn (L.W.); jiangh2015@stu.xjtu.edu.cn (H.J.); 13821175362@163.com (Z.L.); wysun2013195@stu.xjtu.edu.cn (M.M.); male@mail.xjtu.edu.cn (L.M.); 3University Institute of Physical Therapy, University of Lahore, Lahore 54000, Pakistan

**Keywords:** dietary habits, coronary artery disease, cardiovascular disease, dyslipidemia

## Abstract

Background: Adults in South Asian countries have high chances of developing coronary artery disease (CAD) as compared to the developed nations. CAD is among the primary non-communicable causes of death in this region. Dyslipidemia, obesity, smoking hypertension, diabetes are considered as important risk factors for CVD. Methods: A case-control study was conducted, with data was collected from the Punjab Institute of Cardiology in Lahore and the University of Lahore Teaching Hospital. A total of 500 subjects were selected, of which 250 were coronary artery disease patients and 250 were healthy controls. The CAD patients were selected from the outpatient department (OPD) and emergency unit of the Punjab Institute of Cardiology and the University of Lahore Teaching Hospital. Results: The mean age of CAD patients was 57.83 ± 7.51 years and that of the controls was 55.32 ± 6.40 years. There was a significant difference in the mean values of biochemical parameters among cases and controls except for fasting blood sugar levels while there was a significant difference (*p*-value: 0.000) in the mean values of systolic blood pressure among cases and controls. Similarly, the values of diastolic blood pressure were also significantly different (*p*-value: 0.000) among cases and controls. The values of total blood cholesterol, LDL, triglycerides and HDL were also significantly different among cases and controls. There was a significant relationship between consumption of chicken, eggs, beef, yogurt, junk food, fresh vegetables, and fruits, and incidence of CAD. Consuming milk every day, and consuming fish weekly and consuming ghee had no significant association with the risk of coronary artery disease. On the other hand, from the findings of the unadjusted model, there was a significant association between CAD risk and intake of chicken, beef, egg, yogurt, junk food, fish, vegetables, and fruits. Conclusions: Diet is a risk factor for coronary artery disease and can be adjusted to reduce the risk of CAD. A key finding is that consumption of chicken, beef, eggs and junk food are associated with a high risk of CAD whereas consumption of ghee is not associated with the risk of CAD.

## 1. Introduction

The current evidence has shown that globally, the leading cause of death is cardiovascular disease (CVD) and that eighty percent of these deaths occur in under-developed nations. The literature reveals that non-communicable diseases (NCDs) cause sixteen million deaths. Eighty percent of these deaths occur in underdeveloped nations, and, out of these, thirty-seven percent of these deaths are caused by cardiovascular diseases [1,2,3]. Adults in South Asian countries have high chances of developing coronary artery disease (CAD) as compared to the developed nations [4,5] and this is the reason CAD is among the primary cause of death in this region [6]. Dyslipidemia, obesity, smoking, hypertension and diabetes are considered as important risk factors for CVD [7,8].

Studies have reported that one-third of the population’s risk for acute myocardial infarction can be attributed to eating unhealthy foods, such as meat, eggs, and salty snacks. Consumption of foods high in fat and other dietary habits in the West have a significant association with the risk of coronary artery disease [9]. Higher levels of stress, poor eating habits, an inactive lifestyle, and smoking are risk factors that can be modified to prevent CVD [10]. The incidence of CVD is directly related to changes in cholesterol levels [11]. LDL cholesterol levels in young adulthood were also shown to predict the onset of cardiovascular disease later in life [12]. According to Framingham research, smokers have a risk of developing MI and sudden mortality which is proportional to the number of cigarettes smoked each day [13].

Pakistani cuisine is predominantly curry-based and high in saturated fat. The majority of people living in poverty eat carbohydrates and foods high in saturated fats. The findings of a study revealed that the rate of consumption of carbohydrates and fat is fifty one and thirty six percent of the diet, respectively, among the Pakistani population [14]. The prevention of cardiovascular disease is mostly ignored in Pakistan. The authorities devote the majority of their resources to managing communicable diseases and reproductive health issues. For this reason, Pakistan’s health system is unable to deal with the increasing prevalence of non-communicable diseases [15].

This study will create awareness among CAD patients who eventually succumb to the combination of unhealthy dietary habits, smoking, physical inactivity, hypertension, diabetes, hyperlipidemia, hypercholesterolemia. There is a lack of published literature in this region on the association between eating habits and the risk of coronary artery disease. This study was designed to compare the dietary habits of CAD patients to controls matched by gender and age.

## 2. Materials and Methods

### 2.1. Enrollment of Patients

A non-probability convenient sampling technique was used to collect the data for this case control study. The study was ethically approved by the Institutional Review Board Faculty of Allied Health Sciences, The University of Lahore. (IRB no. 879, dated 27 July 2021). The sample size was 246 cases and 246 controls which were rounded off as 250 cases and 250 controls. The following parameters were used to calculate sample size of this case control study: confidence interval 95%, power of the test 80%. The ratio of cases to controls was 1 to 1, the proportion of controls with exposure to unhealthy life styles was 40% while cases with exposure were 53%. [16]. Cases were those patients presenting with one or more CAD risk factors and diagnosed as having angina, STEMI, or NSTEMI based upon ECG, elevated levels of Creatine kinase-MB, and Troponin I. The controls were age and gender-matched healthy people from the same hospital who came with patients as attendants. The CAD patients were selected from the outpatient department (OPD) and emergency unit of the Punjab Institute of Cardiology and the University of Lahore Teaching Hospital.

At the time of admission, the basic risk factors of CAD including age, family history, diabetes, hypertension, overweight, and smoking cigarette were investigated. If one or more of the risk factors were present then as the first step, an ECG of the patients was performed. Based on the ECG findings, blood samples were taken to test levels of Creatine kinase -MB and Troponin I to diagnose CAD, including angina, STEMI, and NSTEMI.

The patients were shifted from the outpatient department or emergency unit to the cardiology ward. The patients were enrolled in the cardiology ward or in CCU based on the diagnosis of either STEMI, NSTEMI, or angina. The data was collected from August 2021 to March 2022.

### 2.2. Data Collection

All study participants gave written informed consent after being told about the study’s scope and objective. A structured questionnaire was used in this study which was pre-tested before data collection from CAD patients and healthy controls. The intravenous blood samples were collected from CVD patients and controls. The response rate among CAD patients was 92%, while that among healthy controls was 96%. There were 20 (8%) of the cases and 10 (4%) of the controls who either refused to participate or left the interview before it was completed.

### 2.3. Anthropometric Measurements

Hydro Weight machine with height measuring stand HF-5664 was used to measure weight. Study participants were advised to remove their shoes and wear light clothing for weight measurement. The weight/height (kg/m^2^) formula was used to measure BMI. The International Obesity Taskforce proposed (IOTF-2000) a cut-off value of BMI for people living in Asian and Indian regions of BMI > 27.50 kg/m^2^, above which respondents were considered obese [17]. The waist circumference was measured in the horizontal plane above the iliac crest, midway between the iliac crest and the lower rib edge [18].

### 2.4. Assessment of Food Habits

A Harvard University food-frequency questionnaire was customized to evaluate diet patterns. This questionnaire refers to 57 food items that are often consumed by people daily, weekly and “during the last 7 days” [19]. The consumption of each food item was measured in terms of the number of servings consumed per day or per week [20].

### 2.5. Clinical and Laboratory Assessment of Coronary Artery Disease

Systolic and diastolic blood pressure was measured by using Certeza Standard aneroid Sphygmanometer CR 1002. Three readings were taken in a row, with a five-minute break between each one. Each subject’s blood pressure was averaged across the three measures. Hypertension was defined as having blood pressure readings of more than or equal to 140/90 mmHg or being on antihypertensive medications [21]. The patients’ history was taken along with the assessment of their medical record to determine if they had hypercholesterolemia, diabetes, or hypertension. The fasting blood sugars were assessed using the glucose–oxidase technique, and the fasting lipid profile (containing cholesterol, LDL, HDL, and triglycerides) was measured using an enzymatic colorimetric approach while Friede Wald’s method was used to determine serum LDL [22]. Serum triglycerides and serum total cholesterol levels of more than 150 mg/dL and more than 200 mg/dL were characterized as hypertriglyceridemia and hypercholesterolemia, respectively, unless the patient was on hyperlipidemia medication. If a respondent’s fasting blood glucose level was higher than 126 mg/dL for two days in a row, or if they had already been diagnosed with diabetes, they were considered as diabetic.

### 2.6. Statistical Analysis

The data was analyzed using SPSS version 24.0. Frequency and percentage were calculated for qualitative variables and mean and standard deviation were calculated for quantitative variables. To compare mean BMI, WC, and WHR between CAD patients and controls, an independent sample t-test was used. Association between study groups and sociodemographic variables was measured using the Chi-square test.

The effect of each food item was estimated using binary logistic regression analysis, and the odds ratio (OR) was obtained using a 95 percent confidence interval. The model was adjusted for age, hypertension, diabetes, BMI and smoking. The *p*-value < 0.05 was taken as significant.

## 3. Results

The mean age of the coronary disease patients was 57.83 ± 7.51 years and that of the controls was 55.32 ± 6.40 years. Among CAD patients there were 52% males and 48% females while among controls there were 51.2% males and 48.8% females. Among CAD patients there were 73 (29.2%) from rural areas and 177 (79.8%) from urban areas, while there were 59 (23.6%) controls from rural areas and 191 (76.4%) from urban areas; there was no significant association in this regard. There were 232 (92.8%) CAD patients who were married while 18 (7.2%) had another status; among controls there were 236 (94.4%) married respondents and 14 (5.6%) had another status. However, a significant difference was seen in the mean values of Body Mass Index, waist-hip ratio, and waist circumference (*p*-values: 0.008, 0.033, and 0.000), respectively. A significant difference in WC and WHR was seen among males (*p*-values: 0.000 and 0.003) whereas no significant difference in WC and WHR was observed among females (*p*-values:0.44 and 0.29) (Table 1).

There was a significant difference in the mean values of biochemical parameters among cases and controls except for fasting blood sugar levels (122.35 ± 25.89 vs. 120.71 ± 26.23, *p*-value: 0.15) while there was a significant difference (*p*-value: 0.000) in the mean values of systolic blood pressure among cases (133.60 ± 4.11) and controls (130.18 ± 2.54). Similarly, the values of diastolic blood pressure were also significantly different (*p*-value: 0.000) among cases (78.64 ± 1.79) and controls (79.44 ± 1.93). The values of total blood cholesterol (200.15 ± 32.49 vs. 177.73 ± 50.47, *p*-value: 0.002), LDL (131.01 ± 30.38 vs. 95.98 ± 18.42, *p*-value: 0.001), triglycerides (263.41 ± 17.66 vs. 169.95 ± 7.07, *p*-value: 0.001) and HDL (38.29 ± 4.27 vs. 44.60 ± 4.69, *p*-value: 0.000) were also significantly different among cases and controls (Table 2).

Among CAD patients the rate of hypercholesterolemia and hypertriglyceridemia was high as compared to controls; these are major risk factors of CAD (*p*-value: 0.000 and 0.000). When comparing CAD patients to controls, total cholesterol, HDL and LDL were significantly different (*p*-value < 0.001, *p*-value < 0.001 and *p*-value < 0.001). The overall lipid profile, systolic and diastolic blood pressure of both CAD patients and controls were greater than the normal cut-off levels. (0.000 *p*-value) A significant association was seen between cigarette smoking and physical inactivity among cases and controls (*p*-values < 0.001 and 0.001) (Table 3).

Binary logistic regression analysis was performed based on the consumption of different food items. In crude analysis, dietary chicken (0.40 (0.243–0.646), *p*-value: 0.000), beef (0.35 (0.216–0.577), (*p*-value: 0.000), egg (0.36 (0.230–0.567), *p*-value: 0.000), yogurt (1.69 (1.054–2.717), *p*-value = 0.029), junk food (French fries, refined bakery products, hamburgers, sweets, sugar drinks, and industrial snacks) (0.22 (0.129–0.366), *p*-value: 0.000), fruits (1.39 (1.254–2.99), *p*-value: 0.003) and vegetables (2.93 (1.815–4.722), *p*-value: 0.000) were associated with the risk of CAD among cases and controls. The model was adjusted for age, hypertension, diabetes, BMI and smoking. After multivariable adjustment for potential confounding factors, a significant association was found for chicken (2.523 (1.54–4.12), *p*-value: 0.000), beef (2.83 (1.73–4.63) *p*-value: 0.000), eggs (2.77 (1.76–4.35), *p*-value: 0.000), yogurt (0.59 (0.37–0.95), *p*-value: 0.027), junk food (4.61 (2.73–7.76), *p*-value: 0.000), fruits (0.52 (0.33–0.79), *p*-value: 0.002) and vegetables (0.34 (0.21–0.55), *p*-value 0.000), respectively, among cases and controls. However, there was no association between cases and controls for consumption of fish less than 2 times per week, milk consumption less than 2 times per week, and consumption of ghee more than 2 times per week (Table 4).

## 4. Discussion

Traditional risk factors including dyslipidemia, and hypertension, are responsible for the development of cardiovascular disease (CVD). Furthermore, these risk factors are dependent upon behavioral factors which include dietary habits and lifestyle [23]. The choice of various food items which we eat may increase or decrease the risk of cardiovascular disease [9]. Most of the researchon the role of dietary habits on cardiovascular diseases is conducted in Europe and the West. The dietary habits of people living in these regions are different from those living in regions like Pakistan; additionally, food cooking methods and preparation of food also vary between different regions and countries. Because of these differences, we have adopted a modified version of the food frequency questionnaire which is a valid and reliable tool to measure daily dietary habits. Food frequency questionnaires were also used in a study called INTERHEART Study [24] which was conducted in fifty-two countries including Pakistan. Due to the social and cultural diversity of people around the world, there is diversity in the eating habits of the people. The findings of a study [25] conducted earlier in Pakistan reported that certain foods (such as meat, eggs and butter) produce protective as well as atherogenic effects) while the findings of our study also reported similar findings as several food items revealed protective as well as atherogenic effects. According to the findings of research conducted in Bangladesh, patients with CAD fall in the category of high risk (BMI > 27.5 kg/m^2^) as compared to the controls [26]. Similar findings have been reported in our study: there were 56.4% of patients belonging to the high-risk group and having a body mass index higher than the cut-off value. BMI and WHR were found to be considerably lower in persons who had good eating habits. These findings support the dietary recommendations suggested by Berg et al. for a healthy weight [27]. Studies [26,28] reported that TC, LDL, HDL, and elevated blood pressure are all closely associated with coronary artery disease. Amani et al. reported that most CAD patients (99 percent) had high levels of triglycerides and LDL-C, whereas almost 50% of the controls reported hypertriglyceridemia and 20% had high LDL levels [29]. The findings of our study were different as in our study high levels of triglycerides and LDL were common among almost half of the CAD patients but for controls, only one-third of respondents had elevated levels of triglycerides and LDL. Rafique et al. reported that in the population of Pakistan, diets with a high frequency of butter, ghee, eggs, and beef has been associated with an increased risk of CAD [25]. Our study reported similar findings as there was a significant association between CAD risk and chicken, beef, eggs, yogurt, and junk food intake, as well as eating fish less than twice per week. Findings of another study [25] revealed that fruit consumption had a protective effect against CAD. These findings are similar to the findings of our study which reported that fruit and vegetable consumption had a protective effect against CAD. The findings of the studies by Framingham et al. and Massachusetts et al. revealed that a healthy eating pattern helps to reduce obesity and maintain BMI and waist circumference at normal levels, both of which are main risk factors for CAD [29,30,31]. Our study findings revealed that there was a strong association between weekly chicken, beef, and egg consumption and daily fruit and vegetable consumption. Similar findings were reported by Shammi et al. [26]. A 27% lower risk of developing CVD was observed among respondents consuming vegetables and fruits three times or more every day [32]. However, the findings of another study [25] reported that greater vegetable consumption had no protective effect against the development of CAD. Our study found no significant association between the risk of CAD and fish consumption less than 2 times per week or consumption of milk and ghee every day. Similar findings were reported by Khatun et al. but their study reported a significant association with weekly consumption of fish which is in disagreement with the findings of our study. This could be due to the fact that in our region fish-eating is specific to certain areas. Giugliano et al. [33] found that a diet high in processed carbohydrates, sugar, saturated and trans fat and lacking in omega-3 fatty acids, natural antioxidants, and fiber from fruits, vegetables, and whole grains affects the immune system, causing inflammation associated with CAD. It is worth mentioning that there has previously been little effort in Pakistan to examine the relationship between red meat and the risk of coronary artery disease [34]. The FFQ’s absence of data on portion sizes is likely a shortcoming of our study. In some other studies, the assessment of dietary intake of CAD patients and controls was done using a questionnaire that measured food intake in grams. Due to this, it is not possible to adjust for the potential confounding influence of nutrients intake in our study.

## 5. Conclusions

Diet is a risk factor for coronary artery disease and can be adjusted to reduce the risk. Consumption of chicken, beef, eggs and junk food are associated with a high risk of CAD whereas consumption of ghee is not associated with the risk of CAD. Consumption of yogurt, vegetables, and fruits, on the other hand, has a preventive effect against coronary artery disease. Anthropometric measures such as BMI, WC, and WHR can also indicate an increased risk of coronary artery disease.

## Figures and Tables

**Table 1 ijerph-19-08635-t001:** Sociodemographic characteristics and anthropometric measurements (*n* = 500).

Variables		CAD Patients	Controls	*p*-Value
Age		57.83 ± 7.51	55.32 ± 6.40	0.35
Gender				
Male		130 (52%)	128 (51.2%)	0.106
Female		120 (48%)	122 (48.8%)	
Residential Area				0.155
Rural		73 (29.2%)	59 (23.6%)	
Urban		177 (79.8%)	191 (76.4%)	
Marital status				
Married		232 (92.8%)	236 (94.4%)	0.46
Other		18 (7.2%)	14 (5.6%)	
Monthly Income				
<20,000		86 (34.4%)	57 (22.8%)	0.12
20,000–40,000		106 (42.4%)	117 (46.8%)	
>40,000		58 (23.2%)	76 (30.4%)	
Body Mass Index		26.87 ± 1.80	25.41 ± 2.04	0.008
WC (cm)	Male	93.74 ± 1.04	90.76 ± 0.97	0.000
WC (cm)	Female	86.23 ± 4.11	85.81 ± 4.16	0.44
WHR	Male	0.95 ± 0.029	0.940 ± 0.019	0.003
WHR	Female	0.93 ± 0.046	0.93 ± 0.035	0.29

Significance was defined as a *p*-value < 0.05.

**Table 2 ijerph-19-08635-t002:** Biochemical parameters associated with risk of CAD among patients and controls (*n* = 500).

Biochemical Measures	Cases (CAD Patients) (Mean ± SD)	Controls (Mean ± SD)	*p*-Value
Systolic blood pressure	133.60 ± 4.11	130.18 ± 2.54	0.000
Diastolic blood pressure	78.64 ± 1.79	79.44 ± 1.93	0.000
Total blood cholesterol (mg/dL)	200.15 ± 32.49	177.73 ± 50.47	0.002
LDL (mg/dL)	131.01 ± 30.38	95.98 ± 18.42	0.001
Triglyceride (mg/dL)	263.41 ± 17.66	169.95 ± 7.07	0.001
HDL (mg/dL)	38.29 ± 4.27	44.60 ± 4.69	0.000
Fasting blood sugar (mmol/L)	122.35 ± 25.89	120.71 ± 26.23	0.15

Significance was defined as a *p*-value < 0.05.

**Table 3 ijerph-19-08635-t003:** Association between cardiovascular disease risk factors and study groups.

Risk Factors	CAD Patients (*n* = 250)	Controls (*n* = 250)	*p*-Value
TG ≥ 150 mg/dL	116 (46.4)	73 (29.2)	0.000
TC > 200 mg/dL	143 (57.2%)	92 (36.8%)	0.000
LDL-C > 100 mg/dL	161 (64.4%)	79 (31.6%)	0.000
HDL-C < 35 mg/dL	167 (66.8%)	71 (28.4%)	0.000
FBS > 126 mg/dL	119 (47.65)	90 (36%)	0.009
BMI > 27.5 kg/m^2^	141 (56.4%)	88 (35.2%)	0.000
BP > 140/90 mm hg	127 (50.8%)	86 (34.4%)	0.000
Cigarette smoking habit	78 (31.2%)	31 (12.4%)	0.000
Physical inactivity	129 (51.6%)	58 (23.2%)	0.000

Significance was defined as a *p*-value < 0.05.

**Table 4 ijerph-19-08635-t004:** Adjusted odds ratio (95% CL) for dietary patterns among coronary artery disease patients and healthy controls.

Food Group	Frequency of Consumption	CAD Patients *n* = 250	Controls *n* = 250	Adjusted Values				Un-Adjusted Values			
				Odds Ratio	95% CI		*p*-Value	Odds Ratio	95% CI		*p*-Value
					Lower	Upper			Lower	Upper	
Chicken	>2 times/week	92 (36.8%)	45 (18%)	2.523	1.54	4.12	0.000	0.40	0.243	0.646	0.000
Beef	>2 times/week	95 (38.0%)	43 (17.2%)	2.83	1.73	4.63	0.000	0.35	0.216	0.577	0.000
Egg	>2 times/week	175 (70%)	132 (52.8%)	2.77	1.76	4.35	0.000	0.36	0.230	0.567	0.000
Fish	<2 times/week	177 (70.8%)	189 (75.6%)	1.19	0.72	1.96	0.50	0.84	0.510	1.390	0.502
Milk	<2 times/week	163 (65.2%)	136 (54.4%)	1.40	0.91	2.16	0.075	0.71	0.462	1.097	0.124
Ghee	>2 times/week	143 (57.2%)	102 (40.8%)	1.32	0.87	2.01	0.18	0.75	0.497	1.155	0.196
Yogurt	>2 times/week	157 (62.8%)	191 (76.4%)	0.59	0.37	0.95	0.027	1.69	1.054	2.717	0.029
Junk food	>2 times/week	91 (36.4%)	29 (11.6%)	4.61	2.73	7.76	0.000	0.22	0.129	0.366	0.000
Fruits	<2 time/day	136 (54.4%)	172 (68.8%)	0.52	0.33	0.79	0.002	1.39	1.254	2.999	0.003
Vegetables	<2 times/day	134 (53.6%)	185 (74%)	0.34	0.21	0.55	0.000	2.93	1.815	4.722	0.000

## Data Availability

Data is available on request from corresponding author.

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
