# Peer review of "Dietary Habits of Patients with Coronary Artery Disease: A Case-Control Study from Pakistan"

_ijerph, 2022, doi:10.3390/ijerph19148635_

Round 1

Reviewer 1 Report

Thank you very much for letting me review this study: "Dietary Habits of Patients with Coronary Artery Disease: A Case-Controlled Study from Pakistan" by Muhammad Kamran Hanif , Le Ma.

The topic is of interest and the study well-conducted. Methods and results are fine and shareable. It is of utmost importance to keep this evidence too. 

I have some suggestions to improve the scientific soundness:

Abstract:

- What does OPD stand for? 

- ghee is associated or NOT to CAD? (see also above)

Introduction:

- the period at lines 55-57 "This study will create awareness among of 37% CAD patients who eventually lead to mortality by unhealthy dietary habits, smoking, physical inactivity, hypertension, diabetes, Hyperlipidemia, hypercholesterolemia. " is pleonastic. I suggest removing it.

Materials and methods

Please, better define cases and controls:

- CAD diagnosis (STEMI, nSTEMI, angina?): what was the admission diagnosis and in what department they were enrolled, when (period from...to..)?

- What kind of patients were the controls? Healthy or other diagnoses? Please be more precise

What does "OPD" stand for? Report entire name before the acronym

Line: 81: (8%) Report number in extenso at the beginning of the period.

Results:

Better define the term "Junk food" and the standard servings

I do not understand if ghee is associated or NOT with CAD. Indeed, in the text you say it is  (Banaspati Ghee more than 2 times increases 1.32 times 181 the chances of developing the CAD t); However, in table 4 the p-value is not significant (0.18 in adjusted, 0,19 in un-adjusted). please check it out.

Reviewer 2 Report

Dear Authors,

The results presented in the text [lines 128-140] with reference to Table 2 do not correspond to the results contained in Table 2.

“A significant difference was reported in the mean values of systolic (133.60 ± 4.11 vs 130.18 ± 2.54, p-value: 0.000) and diastolic blood pressure (78.64 ± 1.79 vs 79.44 ± 0.000, p- value: 0.000), total cholesterol (200.15 ± 32.49 vs 177.73 ± 50.47, p-value: 0.002), LDL (131.01 ± 30.38 vs 44.60 ± 4.69, p-value: 0.000 ), triglycerides (263.41 ± 17.66 vs 169.95 ± 7.07) and HDL (38.29 ± 4.27 ± 25.89 vs ±120.26 ± 26.23, p-value: 0.000) among CAD patients in comparison to matched healthy controls. On the other side, no significant difference can be seen in the mean values of fasting blood sugar among CAD patients and Controls. (122.35 ± 25.89 vs 120.71 ± 26.23, p-value: 0.15) (Table 2).”

The results presented in the text [lines 162-186] with reference to Table 4 do not correspond to the results contained in Table 4.

Yours sincerely

Round 2

Reviewer 1 Report

The manuscript improved.

I have only a few more minor remarks:

You wrote that "Consumption of Ghee more than 2 times per week is not associated with the risk of developing CAD. (Corrections have been made on page 6, lines 196-197)"

This information should also be corrected in the abstract: indeed in the last period it is written that “ ghee is associated with a high risk of CAD.”

---

Response 8: Added definition (page 6, line 186)

"Junk foods include pasta, pizza, burger, unhealthy sandwiches, etc.

Standard Serving is the amount of food that is generally served."

RE: sorry but this is misleading information: pasta and pizza aren’t junk food. Pasta (especially of whole grain) is the basis of the Italian and Mediterranean diets. And Italian pizza is a wonderful meal (even if it cannot be eaten every day). So, please, replace them with “French fries, refined bakery products, hamburgers, sweets, sugar drinks, and industrial snacks”

Best regards,
